# Synthesis of PMMA Microspheres with Tunable Diameters: Evaluation as a Template in the Synthesis of Tin Oxide Coatings

**DOI:** 10.3390/polym15112419

**Published:** 2023-05-23

**Authors:** José L. Mendoza-Castellanos, Juan C. Pantoja-Espinoza, Luis C. Rodríguez-Pacheco, Francisco Paraguay-Delgado

**Affiliations:** Centro de Investigación en Materiales Avanzados S. C. (CIMAV), Laboratorio de Síntesis de Óxidos Semiconductores, Departamento de Física de Materiales, Miguel de Cervantes 120, Complejo Industrial Chihuahua, Chihuahua 31136, Mexico; luis.mendoza@cimav.edu.mx (J.L.M.-C.); luis.rodriguez@cimav.edu.mx (L.C.R.-P.)

**Keywords:** PMMA spheres, macroporous, coatings, tin oxide

## Abstract

The synthesis of polymethyl methacrylate (PMMA) spheres with different sizes has been a challenge. PMMA has promise for future applications, e.g., as a template for preparing porous oxide coatings by thermal decomposition. Different amounts of SDS as a surfactant are used as an alternative to control PMMA microsphere size through the formation of micelles. The objectives of the study were twofold: firstly, to determine the mathematical relationship between SDS concentration and PMMA sphere diameter, and secondly, to assess the efficacy of PMMA spheres as templates for SnO_2_ coating synthesis and their impact on porosity. The study used FTIR, TGA, and SEM techniques to analyze the PMMA samples, and SEM and TEM techniques were used for SnO_2_ coatings. The results showed that PMMA sphere diameter could be adjusted by varying the SDS concentration, with sizes ranging from 120 to 360 nm. The mathematical relationship between PMMA sphere diameter and SDS concentration was determined with a y = ax^b^ type equation. The porosity of SnO_2_ coatings was found to be dependent on the PMMA sphere diameter used as a template. The research concludes that PMMA can be used as a template to produce oxide coatings, such as SnO_2_, with tunable porosities.

## 1. Introduction

Polymethyl methacrylate (PMMA) is widely used in electronic equipment, medical technology, and polymer membrane areas [1]. In addition, some authors have described PMMA as a template for metallic oxide synthesis to improve properties such as pore size distribution and superficial area [2]. Hyodo et al. used PMMA microspheres at 150, 250, 400, and 800 nm to synthesize macroporous SnO_2_ films to improve NO_x_ and hydrogen (H_2_) detection [3]. Furthermore, Kamitani et al. used 98.6 µm microspheres to make high porosity alumina films for possible uses as filters, insulators, and catalytic support [4]. Another interesting work is that of Khan et al. who, with spheres of approximately 300 nm diameter, fabricated a porous film of TiO_2_ in heterojunction with ZnO for potential use as an efficient photoanode in a photoelectrochemical cell [5]. It is also important to mention the research of Sordello and Minero, who synthesized microspheres with 250 nm diameter porous TiO_2_ powders with Pt to improve the H_2_ production of this semiconductor [6].

Tin oxide (SnO_2_) coatings have gained significant attention due to their remarkable properties and potential applications in various research fields [7]. The properties of SnO_2_, such as high transparency, high refractive index, high chemical stability, and excellent electrical conductivity, make it an attractive material for use in a wide range of technology applications. These applications include gas sensors [8], solar cells [9], photocatalysts [10], electrochromic devices [11], and antistatic coatings [12]. The versatility and flexibility of SnO_2_ coatings have made them a popular research topic, with numerous studies being conducted to explore their various applications [13]. The future of SnO_2_ coatings is bright, with potential for further research and development to expand their applications and improve their properties [14].

The polymer PMMA is a transparent thermoplastic synthesized by emulsion polymerization, solution polymerization, and bulk polymerization from the MMA (methyl methacrylate) monomer [15,16]. Emulsion polymerization is one of the most commonly used methods to synthesize PMMA owing to its high reactivity and property of simultaneous increase in the polymerization rate and molecular weight [1]. This method uses a surfactant, a monomer, and an initiator [1,2].

Some initiators as tertiary amines, such as 2-(N, N-dialkylamino)ethanol and N, N, N′, N′-tetramethyl ethylenediamine, exhibit a higher initiation activity during PMMA polymerization [17]. In addition, H_2_O_2_ [10] and potassium peroxy-disulfate [18] have been reported as initiators. However, the most common initiators used for PMMA synthesis are ammonium persulfate (APS) [19,20,21,22] or APS/ascorbic acid mixtures [23] that act as redox initiators.

Various factors impact the PMMA synthesis process. Among these, the utilization of surfactants is often explored in existing literature due to their impact on PMMA’s morphology, particle size variation, and particle shape. Parra et al. used anionic (sodium lauryl sulfate, SLS) and nonionic (nonylphenol ethoxylated) surfactants and reported a high dependence of the nature and concentration of the surfactant on the morphology and particle size, with the anionic surfactant SLS being better than the non-ionic surfactant [24]. Baissac et al. prepared PMMA under different time and power parameters and small amounts of the surfactants Triton X-100 and Tween 20 and studied their effect on particle diameters between 78 and 310 nm [25]. Mahmoudian et al. used Triton X-100 as a surfactant for PMMA synthesis and reported sphere diameters between 100 and 300 nm [26]. Vargas-Salazar et al. synthesized PMMA by ultrasound at 50 °C with Hitenol BC10 as a surfactant and reported average particle diameters between 39 and 63 nm [27]. However, the most frequently used surfactant for PMMA synthesis is sodium dodecyl sulfate (SDS), for which some reports have found particle diameters between 20 and 97 nm [28] and between 100 and 1000 nm [29].

Sodium dodecyl sulfate can form micelles in water, which can adsorb onto the surface of the growing PMMA particles and prevent them from coagulating or aggregating [30]. SDS can also lower the interfacial tension between the water and the monomer phases, facilitating the formation of smaller and more uniform droplets [31].

The role of SDS in the formation of PMMA microspheres depends on several factors, such as the concentration of SDS, the monomer concentration, the initiator concentration, the reaction temperature, and the stirring speed [30,31,32]. The optimal conditions for obtaining monodisperse PMMA microspheres with high yield and stability vary depending on the desired size and morphology of the microspheres [30,31,32].

Few works report PMMA sphere synthesis with different diameter sizes depending on the variation of the amount of surfactant. Kamras et al. prepared PMMA by emulsion polymerization using microwave radiation as a heating source. They measured the Z-average particle size using the dynamic light scattering (DLS) technique, reporting particle sizes between 97.36 and 23.19 nm using 0.008 and 0.2 % SDS [28]. There has been no report about PMMA sphere synthesis by emulsion polymerization and conventional heating as a function of the variation of the amount of SDS as surfactant. In addition, the effect of PMMA sphere size as a template on SnO_2_ porosity synthesis has not been reported.

Therefore, the objectives of this research were twofold: first, find a mathematical function that expresses the change in the size of PMMA sphere diameter as a function of the surfactant SDS concentration. Second, evaluate PMMA spheres as a template and their effect on porosity for SnO_2_ coating synthesis.

## 2. Materials and Methods

### 2.1. Synthesis of PMMA

The synthesis method for all PMMA spheres is by batch emulsion polymerization [19,33,34,35]. The followed reagents were used: sodium dodecyl sulfate (SDS) (Sigma-Aldrich, 99%, Tokyo, Japan) as a surfactant; methyl methacrylate (MMA) (Aldrich, 99%, St. Louis, MI, USA) as a monomer, and ammonium persulfate (APS) (Sigma-Aldrich, 98%, Japan) was used as initiator.

Figure 1a shows the experimental arrangement system used in the polymerization reaction to obtain PMMA spheres. First, 150 mL tri-distilled water (J. T. Baker, México) was placed with stirring at 400 rpm in a 3-hole flask. An inert argon atmosphere was maintained at a flow rate of 100 mL/min (Figure 1a) to maintain the inert atmosphere. Under these conditions, the tri-distilled water was heated using a heating mantle (Figure 1b) up to 65 °C and maintained by the temperature controller (Figure 1c) with ±1 °C. Following this, 0.0055 g of SDS was dissolved in 10 mL of tri-distilled water and pre-heated at 65 °C on the heater (hot plate); this last solution was added to a 3-hole flask and maintained with stirring for 20 min. Then, 42 mL of the MMA monomer, previously heated on the hot plate at 65 °C, was added to the solution (3-hole flask); the resultant dissolution was maintained with stirring for 30 min, obtaining a clear solution (Figure 1b). Following this, the polymerization initiator was prepared by dissolving 0.3434 g of APS in 10 mL of tri-distilled water.

Finally, the APS solution was slowly dosed to initiate the polymerization reaction. At this time, the polymerization process began; after 1 h, the transparent solution changed to a white color as observed in Figure 1d,e. Then, this PMMA-spheres dispersion was cooled to room temperature, and the rest of the reagents were separated by centrifuge at 5000 RPM for 1 h; this washing process was made in tri-distilled water 5 times continuously. The resulting product was dried at 70 °C for 12 h until a white powder was obtained. The sample was identified as 0.0139, which corresponds to the SDS surfactant percentage concentration concerning the MMA monomer. The above procedure was repeated in triplicate with the same synthesis method using the following weights of SDS 0.0055, 0.0083, 0.011, 0.0138, 0.0165, 0.0275, and 0.55 g of surfactant; then the obtained samples were identified as 0.0139, 0.0210, 0.0279, 0.0349, 0.0418, 0.0696, and 0.1391; these values represent the different SDS % value concentrations.

### 2.2. Synthesis of Porous Tin Oxide Coatings

PMMA spheres as a template were used to synthesize SnO_2_ coatings with different pore sizes according to the previously reported procedure [36]. In brief, the precursor dispersion was made using PMMA microspheres as templates; the precursor salt of SnO_2_ was tin (IV) tetrachloride pentahydrate with 98% purity purchased from the Sigma-Aldrich company. Then, the dispersion was deposited on a glass substrate by the doctor blade method and dried in a closed system at room temperature. Then, the sample was underwent heat treatment at 400 °C, and it was applied to decompose the microspheres of PMMA to obtain porous SnO_2_ coatings (Figure 2). Finally, the coatings’ structure and morphology were characterized with scanning and transmission electron microscopy (SEM and TEM, JEOL) techniques for all obtained coatings with different porous diameters.

### 2.3. Characterization

The PMMA microspheres were characterized to determine their purity. First, Fourier Transform Infrared Spectroscopy (FT-IR, Schimadzu IRAffinity-1S, Kyoto, Japan) in transmittance mode with an ATR accessory was used to acquire spectra at room temperature from 4000 to 400 cm^−1^ to identify the characteristic functional groups of the obtained samples. From the thermogram obtained by TGA was determined the decomposition temperatures (TGA-SDT Q600, New Castle, Delaware, USA). The morphology of PMMA microspheres was observed with a scanning electron microscope (SEM, JEOL Ltd., JSM7401F, Tokyo, Japan). The structure of SnO_2_ obtained was determined by X-ray diffraction and electron diffraction (Panalytical, Almelo, Holland).

The morphology of the SnO_2_ coatings was studied by transmission electron microscope (TEM, JEOL Ltd., JEM 2200FS+Cs, Tokyo, Japan) and SEM (Hitachi SU3500, Tokyo, Japan) techniques. The selected area electron diffraction (SAED) technique confirmed the crystalline structure. In addition, from the SEM micrographs were measured the pore diameters using Gatan Digital Micrograph(r) Software (version 3.7.0., Pleasanton, CA, USA).

## 3. Results and Discussion

### 3.1. FTIR Analysis

The FTIR spectrum of the PMMA spheres recorded by ATR mode is shown in Figure 3. Three characteristic peaks of PMMA appear at 1062, 985, and 845 cm^−1^ [37]. The bands observed at 1387 and 750 cm^−1^ correspond to the α-CH_3_ group [38]. The absorption bands centered at 1483, 1445, and 1435 cm^−1^ are associated with symmetrical deformation vibrations of the CH_3_ group [19]. The CH_2_/CH_3_ stretching modes are at 2994 cm^−1^ and 2950 cm^−1^; the C=O stretch is at 1725 cm^−1^ from the C-COO group, and the asymmetric C–O–C stretch is at 1243, 1194 cm^−1^, and 1144 cm^−1^. These vibrations are visible for synthesized PMMA spheres [39]. Different quantities of SDS as a surfactant did not affect the chemical composition of PMMA; the SDS surfactant used during the emulsion polymerization process was removed through multiple washings and centrifugations. Therefore, it is unlikely that there would be any significant SDS peaks present in the FTIR spectra of the resulting PMMA product.

### 3.2. TGA Analysis

The TGA thermogram of the synthesized PMMA material by emulsion polymerization is shown in Figure 4. 

It reveals a thermal decomposition process in a single step. According to the TGA sample behavior, it was separated into two regions, as follows. (i) First step: the mass loss for the PMMA samples remains unchanged up to 286 °C, where it starts to decompose. (ii) Second step: the PMMA decomposes into monomers between 286 and 400 °C [40,41], ending with the CO_2_ and CO formation [42]. It can be deduced that the PMMA obtained is a thermally stable polymer [19], and the temperature needed to decompose the spheres to form the tin oxide coating must be set above 400 °C. The amount of SDS used in the emulsion polymerization process does not affect the decomposition temperatures observed in the TGA analysis because the SDS is removed from the final product through washing and centrifugation. The TGA analysis measures the thermal stability of the polymer, and the SDS does not contribute to this property. Therefore, the amount of SDS used in the emulsion polymerization process does not affect the TGA results.

### 3.3. Morphology by SEM 

Figure 5 shows SEM images of PMMA nanoparticles for all synthesized samples. It can be seen that the aspect for all synthesis conditions is completely spheric.

The sphere diameter decreases as the surfactant quantity concentration increases, with uniform size. These images confirm the formation of spherical micelles for this range of surfactant concentrations. This means the micelles are arranged with small sizes at higher concentrations, remaining spherical. In addition, it is possible to observe some particle agglomerates with different sizes.

Figure 6 shows the statistical distribution of the diameter measurement values of PMMA spheres.

The quantitative data corresponding to the sphere sizes were obtained from acquired SEM images (Figure 5). Figure 6a shows the distribution of PMMA sphere sizes, starting with the lowest surfactant concentration up to the higher SDS content. These average diameters of PMMA sphere sizes decreased systematically as the SDS concentration increased. These values were 360 ± 14 265 ± 16, 245 ± 15, 216 ± 16, 178 ± 8, 156 ± 10, and 120 ± 8 nm for the concentrations of 0.0139, 0.0210, 0.0279, 0.0349, 0.0418, 0.0696, and 0.1391% of SDS content, respectively. The lowest standard deviation of measurement values confirm their uniformity, as can be noticed in the SEM images. These diameter values are bigger than those obtained by [43]; they find diameter values between 20 and 100 nm at a similar surfactant concentration. This confirms the small spherical micelle formation in the synthesis procedure. Our procedure aims to find a spheres diameter between 120 and 360 nm, and this interval size is helpful for different technological applications.

Table 1 shows the relationship between surfactant concentration and sphere diameter in previous studies and in our own. The data shows that increased surfactant concentration leads to decreased sphere diameter. This trend is consistent with our results, where the size of the spheres decreased as the amount of SDS increased. These findings demonstrate the importance of surfactant concentration in controlling the size of the spheres and, ultimately, the pore size in coatings, which has important implications for various applications.

Some studies have shown that increasing the concentration of SDS can lead to smaller and more spherical PMMA microspheres and lower conversion and higher polydispersity [30,32]. Other studies have reported that increasing the concentration of SDS can increase the conversion and decrease the polydispersity but also increase the size and reduce the sphericity of the PMMA microspheres [31]. These discrepancies may be due to different experimental conditions and methods researchers use [30,31,32]. This is the case of the information in Table 1, where different synthesis techniques led to a different PMMA sphere size.

The interaction between SDS and PMMA microspheres or micelles can be explained by two models: (1) the electrostatic stabilization model, which assumes that SDS micelles carry a negative charge that repels other micelles or particles; and (2) the steric stabilization model, which assumes that SDS micelles form a layer around the particles that prevents them from coming into contact with each other [30]. Both models may contribute to the phenomenon observed in the works presented in Table 1, as well as the results of this article, where an increase in SDS causes a decrease in sphere diameter size.

### 3.4. Fit a Function 

Figure 7 shows the PMMA sphere diameter size changes as a function of SDS concentration. The behavior between these variables showed an exponential relationship, which was fitted using the Levenberg–Marquardt algorithm, given by the general Equation:y = ax^b^,(1)

The fit of the function relating the diameter of PMMA spheres to the SDS surfactant concentration is shown as follows: PMMA diameter (nm) = 37.99x^−0.5168^,(2)

Table 2 shows the parameters of the fitted function to relate the change in PMMA sphere diameter as a function of SDS surfactant concentration.

The results of this investigation are similar to those found by other authors on the change of PMMA sphere size as a function of SDS surfactant concentration, in which they report a nonlinear behavior, although with a second-order exponential decay function [28]. Although surfactant concentration was similar, the particle size intervals show considerable differences in nanoparticle sizes (20–97 nm) relative to this work. Possible causes of the differences in particle size ranges may be factors such as the type of conventional or microwave heating, the polymerization temperatures at 65 or 90 °C, or the different initiators used, such as APS or potassium persulfate (KPS).

Table 3 shows the PMMA sphere diameters values obtained from the statistical distribution measurements on the micrographs and the diameters calculated from the nonlinear function.

The table shows that the calculated PMMA diameters are within the minimum and maximum ranges of each mean value obtained for each concentration of the SDS surfactant. Regarding 0.0418% SDS, the particle diameter calculated falls outside the minimum and maximum value of the measured value. The equation modeling the diameter size of PMMA spheres as a function of % SDS is valid between 120 to 360 nm. Kamras et al. reported the validity of a second-order exponential decay equation between 20 and 97 nm [28]. Other works reported a relationship with particle size between 300 and 1000 [28,44]. This fitting shows a good mathematical function that is predictable for the diameter parameters tunability synthesis.

### 3.5. Synthesis of SnO_2_ Coatings

Figure 8 shows SAED patterns of the SnO_2_ particles (Figure 8a) and the brightfield TEM images of the SnO_2_ coating under different magnifications (Figure 8b–d). Figure 8a shows the selected area electron diffraction (SAED) patterns of SnO_2_ nanoparticles prepared with a PMMA microspheres template. The rings correspond to the (1 1 0), (1 0 1), (2 0 0), and (2 1 1) d-spaces; they belong to the tetragonal phase of SnO_2_, similar to the XRD pattern.

These rings agreed with the XRD intensities previously reported, and the SAED patterns pattern indexed to SnO_2_ [41] too. Micrographs with different scale sizes show a macroporous morphology and pore diameter of 262 nm. The macroporous size of SnO_2_ coating correlated with the average microsphere size of PMMA used for synthesis. This porosity is due to the decomposition of the PMMA microspheres template by heat treatment at 400 °C which was previously synthesized using the surfactant under the concentration of 0.0279 % SDS, obtaining an average diameter of 245 nm.

Figure 9 shows SEM images of SnO_2_ coatings morphology with different-sized porous, generated using different PMMA spheres as a template. 

The micrographs were obtained at 10,000 × times magnifications with 5 µm scales. Figure 9a corresponds to the SnO_2_ coating synthesized using the PMMA spheres diameter of 360 nm (0.0139% SDS), in which the porous morphology can be clearly seen. The statistical distribution shown in Figure 9b corresponds to the pore size measurements giving an average of 342 nm, which is a pore size caused by the spaces generated due to the decomposition of PMMA with a sphere diameter size of 360 nm. The pore size started to decrease, as shown in the micrographs in Figure 9c,e,g, maintaining the expected behavior since PMMA spheres were used as templates with sphere diameters of 245, 216, and 120 nm synthesized using the concentrations of 0.0279, 0.0349, and 0.1391% SDS as a surfactant, respectively. The statistical pore size distributions corresponding to these micrographs are shown in Figure 9d,f,h, in which the frequency shift to the left can indicate the decrease in pore size consistent with the porosity observed in the micrographs. The average pore sizes obtained were 238, 212, and 91 nm, and likewise, it can be affirmed that the porosity size in the SnO_2_ coatings was due to the thermal decomposition of the PMMA spheres formed by the different diameter sizes.

Table 4 shows a summary of pore sizes of different materials synthesized using PMMA as a template; the pore sizes of the SnO_2_ coatings obtained in this work are comparable with the sphere size of PMMA used as a template reported in the literature, regardless of the synthesis conditions for their preparation. This work highlights, in addition to the synthesis of PMMA with controllable sphere size, the synthesis of SnO_2_ coatings with controllable pore size.

Figure 10 shows the relationship between the porosity of SnO_2_ coatings with the diameters of PMMA spheres obtained experimentally under the effect of surfactant concentration. In addition, the relationship of the pore size of SnO_2_ coatings with the PMMA sphere diameter obtained from the mathematical fitted function to the experimental data is shown. It can be seen in Figure 10 that the pore diameter of the coatings decreases as the PMMA sphere size decreases, and that this has a relationship with the SDS surfactant concentration quantity, whose diameter variation has been studied in this work. It is again demonstrated that the function is also exponential.

## 4. Conclusions

The effect of SDS surfactant on the synthesis of PMMA with different sphere diameters was studied and evaluated as a template for obtaining SnO_2_ coatings with different porosities. The PMMA sphere diameters between 120 and 360 nm were synthesized using different amounts of surfactant in the range of 0.0139 and 0.139 % SDS with a conventional radical polymerization technique. For size-tuning of PMMA sphere diameter as a function of % SDS was fitted to the mathematical equation y = ax^b^. The pore size of SnO_2_ coatings varies as a function of the PMMA-sphere diameter used as a template. This research demonstrates that it is possible to obtain oxide coatings such as SnO_2_ with size-tunable porous using PMMA as a template. The wall of these macropores is built with nanometric-size SnO_2_ particles. Similar coating porosities could be synthesized using PMMA spheres with different oxides for different technological applications.

## Figures and Tables

**Figure 1 polymers-15-02419-f001:**
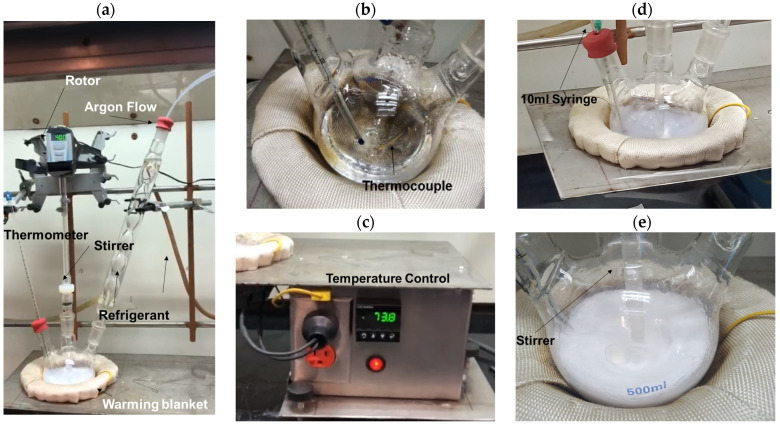
PMMA sphere synthesis by emulsion polymerization: (**a**) overview, (**b**) temperature controller, (**c**) thermometer and thermocouple, (**d**) stirrer, and (**e**) PMMA.

**Figure 2 polymers-15-02419-f002:**
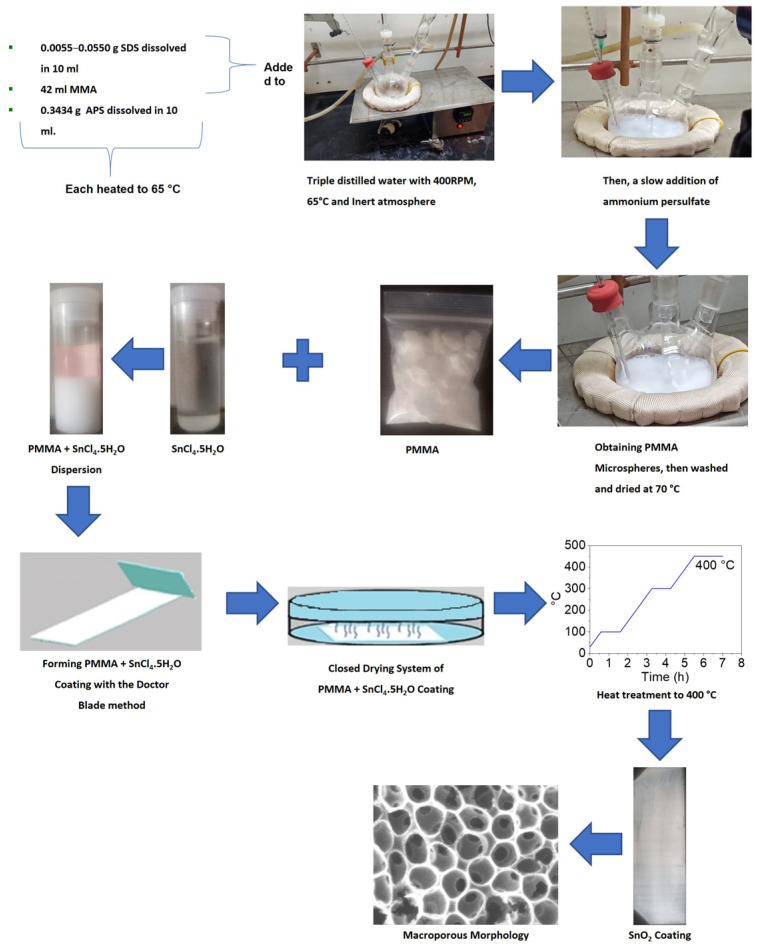
General scheme, starting with the PMMA sphere synthesis as a template and ending with the synthesis of porous SnO_2_ coatings.

**Figure 3 polymers-15-02419-f003:**
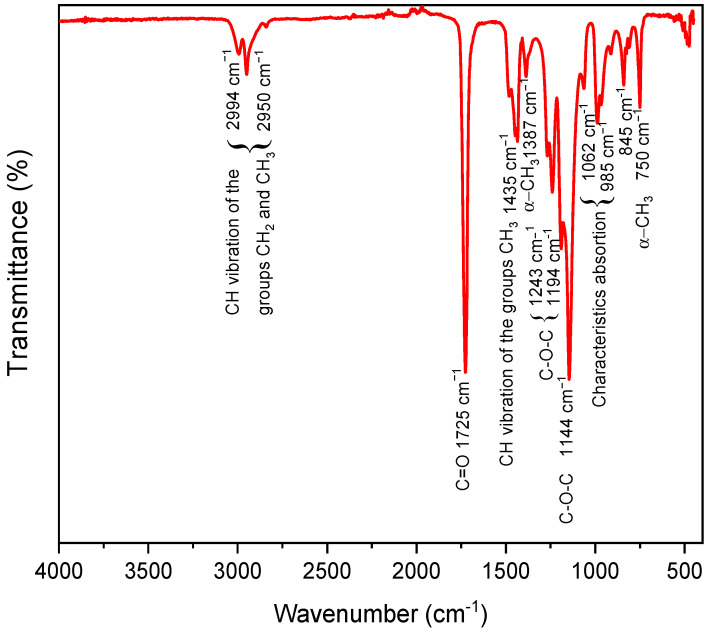
FTIR spectrum acquired from the synthesized PMMA spheres.

**Figure 4 polymers-15-02419-f004:**
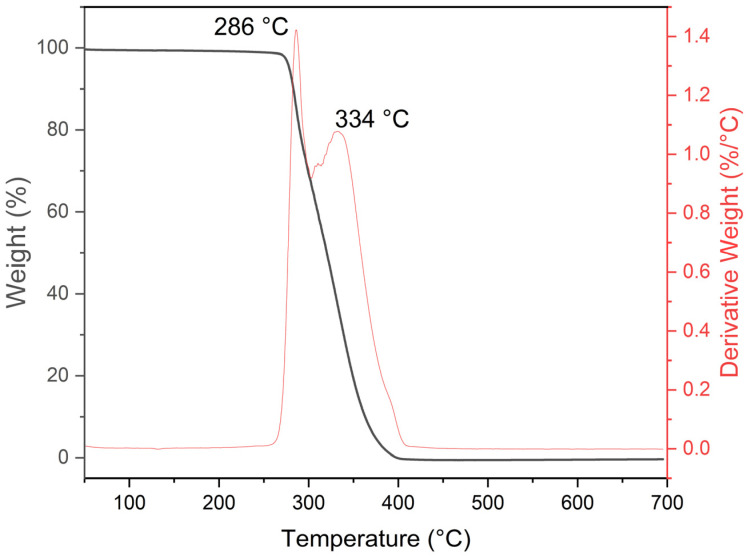
TGA thermogram showing decomposition of PMMA.

**Figure 5 polymers-15-02419-f005:**
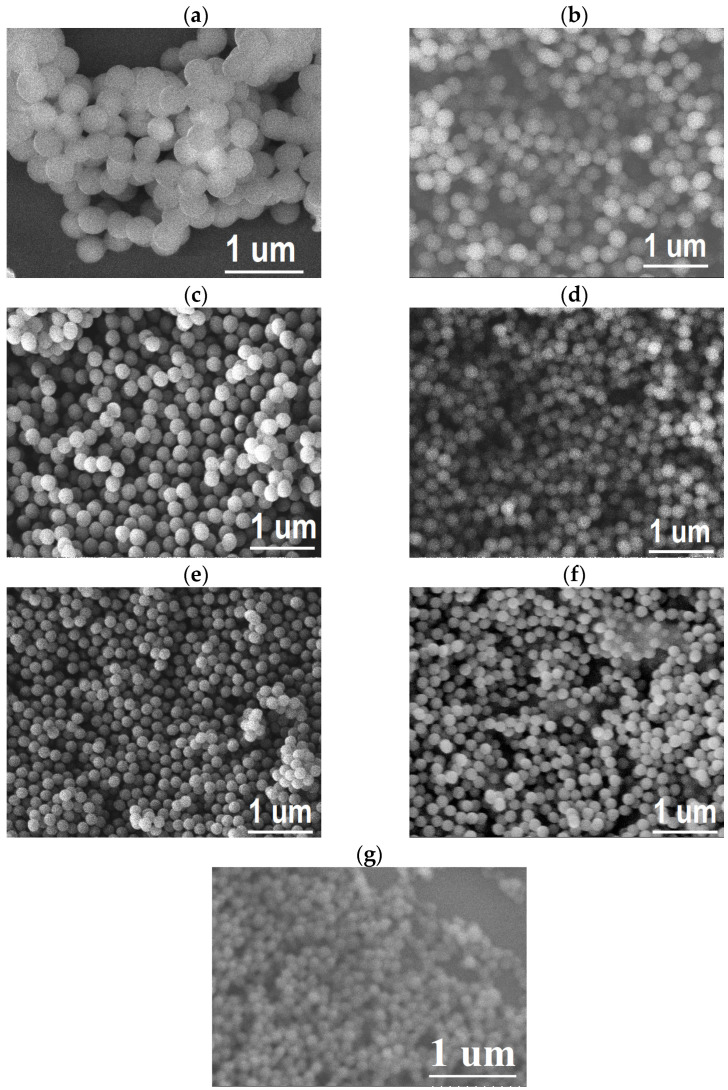
SEM images of PMMA spheres for (**a**) 0.0139, (**b**) 0.0210, (**c**) 0.0279, (**d**) 0.0349, (**e**) 0.0418, (**f**) 0.0696, and (**g**) 0.1391% SDS content.

**Figure 6 polymers-15-02419-f006:**
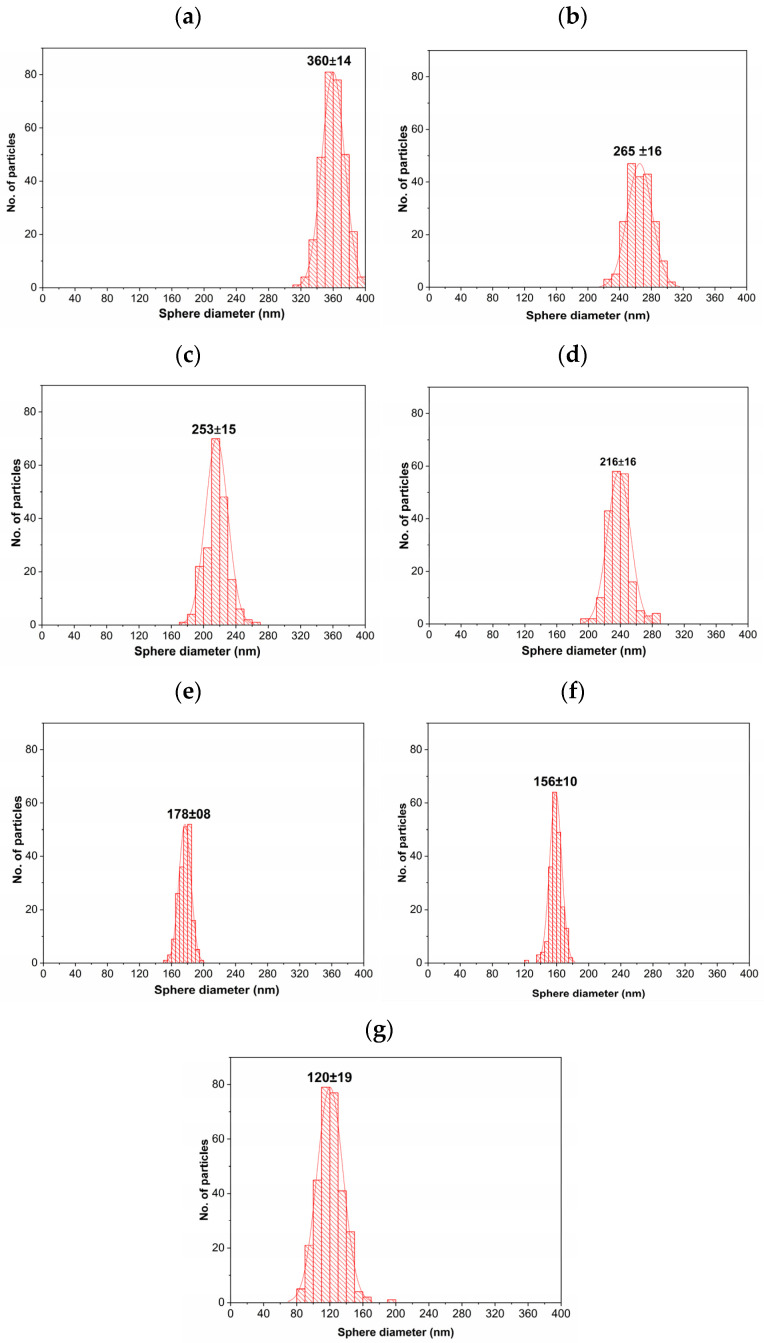
Statistics distribution of PMMA sphere diameter for (**a**) 0.0139, (**b**) 0.0210, (**c**) 0.0279, (**d**) 0.0349, (**e**) 0.0418, (**f**) 0.0696, and (**g**) 0.1391% SDS content.

**Figure 7 polymers-15-02419-f007:**
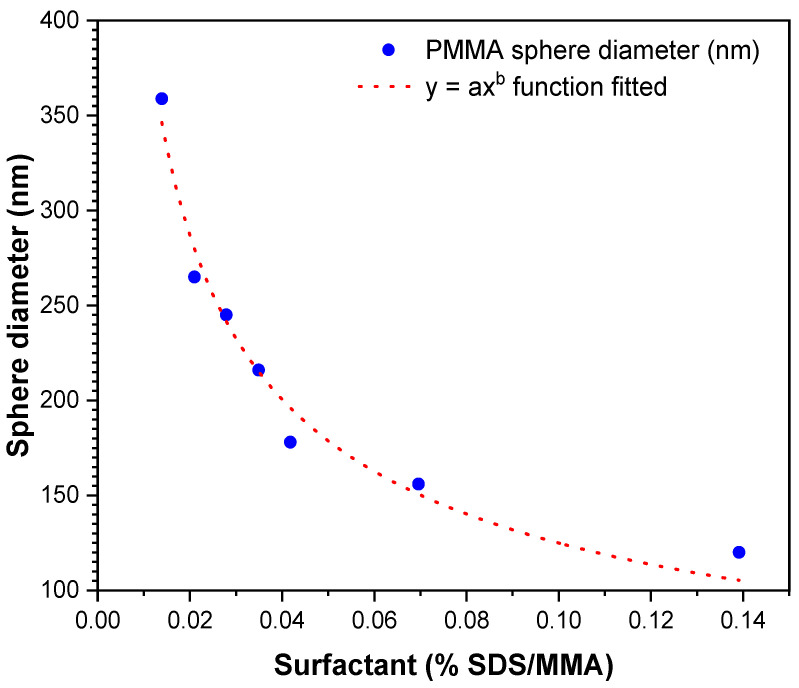
Diameter size fitting as a function of surfactant shows tunability of PMMA sphere diameter.

**Figure 8 polymers-15-02419-f008:**
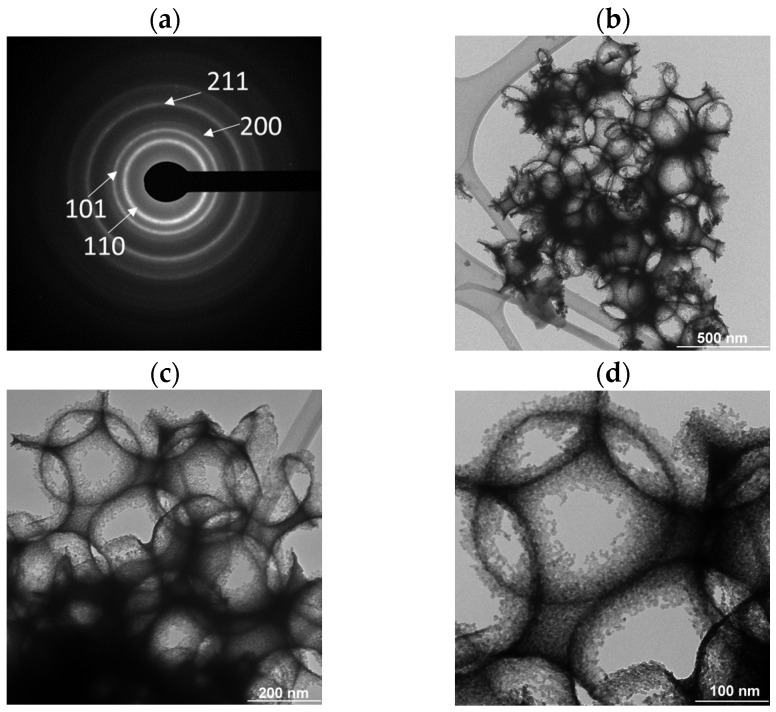
(**a**) SAED pattern and brightfield TEM images with (**b**) 500 nm, (**c**) 200 nm, and (**d**) 100 nm scale bars.

**Figure 9 polymers-15-02419-f009:**
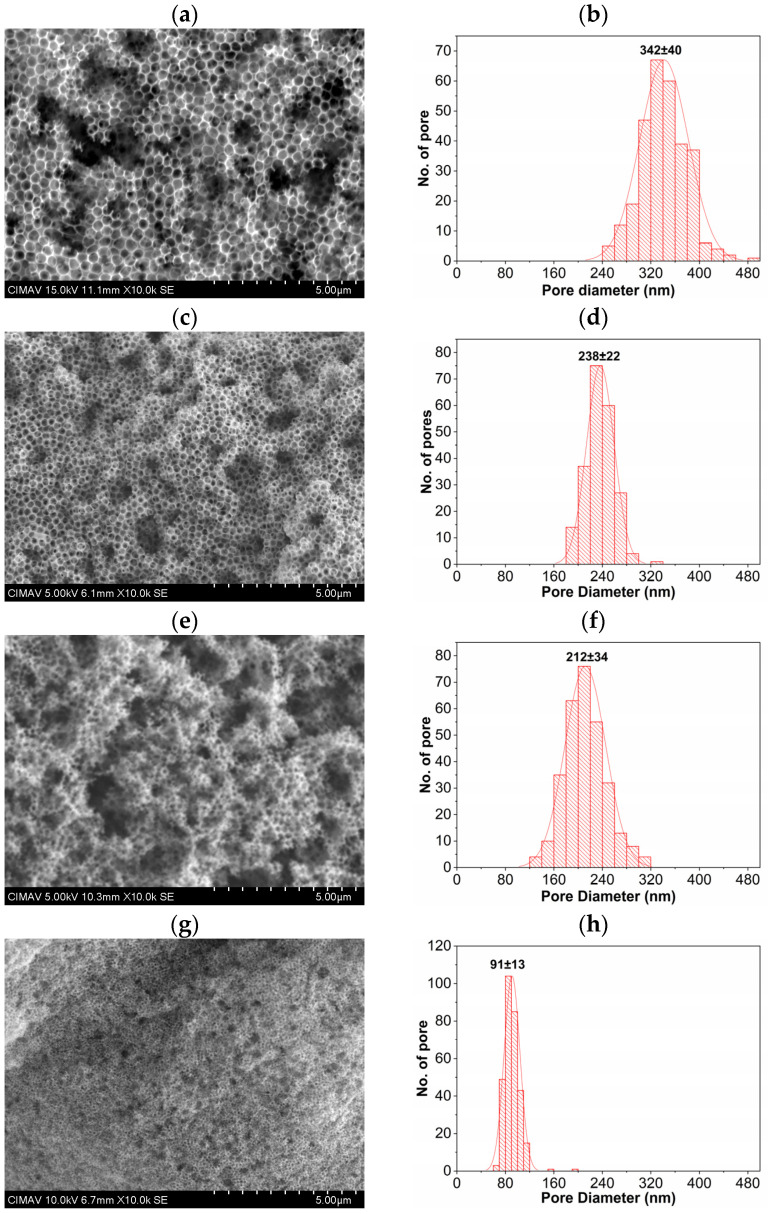
SEM images of SnO_2_ coatings (**left**) and statistical distribution of pore sizes (**right**) prepared with PMMA spheres as a template obtained with different % of SDS content: 0.0139 (**a**,**b**), 0.0279 (**c**,**d**), 0.0349 (**e**,**f**), and 0.1391 (**g**,**h**).

**Figure 10 polymers-15-02419-f010:**
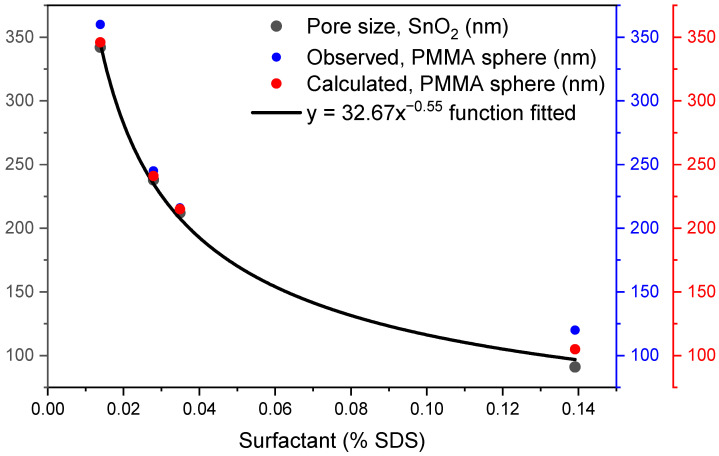
Comparison of the pore size of SnO_2_ coatings (black) with the diameter of observed (blue) and calculated (red) PMMA spheres.

**Table 1 polymers-15-02419-t001:** Summary reported activities for PMMA sphere synthesis using SDS as surfactant.

Surfactant	Surfactant (%)	Sphere Diameter (nm)	Synthesis Method	Reference
SDS	0.008–0.2	97–23	Microwave heating	[28]
SDS	0.09–0.017	60–20	Magnetic stirring and ultrasound	[43]
SDS	0.0139–0.1391	360–120	Emulsion polymerization	This work

**Table 2 polymers-15-02419-t002:** Parameters model in fitted function.

Parameter	Value
Equation	y = ax^b^
a	37.99 ± 5.75
b	−0.5168 ± 0.0406
R^2^	0.9746

**Table 3 polymers-15-02419-t003:** SEM-analysis vs. formula-calculated sphere diameter of PMMA.

SDS (%)	PMMA Sphere Diameter (nm)	
	Measured SEM Analyses	Calculated
0.0139	360 ± 14	346.31
0.0210	265 ± 16	279.80
0.0279	245± 15	241.59
0.0349	216 ± 14	215.19
0.0418	178 ± 08	196.03
0.0696	156 ± 10	150.62
0.1391	120 ± 19	105.30

**Table 4 polymers-15-02419-t004:** Summary values of porous materials synthesized using PMMA as a template.

Material	Sphere Diameter (nm)	Pore Size (nm)	Ref.
TiO_2_	87	87	[45]
ZnFe_2_O_4_	291	203	[46]
TiO_2_	250 420	125 220	[6]
CeO_2_	325	240	[47]
WC	490 180	371 149	[48]
KTiOPO_4_/SiO_2_	250	200	[49]
LaFeO_3_	270	191	[50]
SnO_2_	360	342	[36]

## Data Availability

The data presented in this study are available on request from the corresponding authors.

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
