# Peer review of "Synthesis of PMMA Microspheres with Tunable Diameters: Evaluation as a Template in the Synthesis of Tin Oxide Coatings"

_polymers, 2023, doi:10.3390/polym15112419_

Round 1

Reviewer 1 Report

This paper investigated the effect of surfactant ratio on the diameter of PMMA spheres and SnO2 coating was also reported. However the comments below should be addressed before publication.

Some sentences need revision including but not limited to line 49, 249-242.

Typos need to be fixed including but not limited to porosity materials.

Experimental design should be better explained. How are the surfactant/monomer ratios determined? What happens if lower or higher concentrations are used ? What are the limits? 

The discussions are poor especially for SEM images. Authors only presented the results and there is not enough discussion on the results. 

There are similar studies however the discussions and novelty is not presented clearly. The table should be provided and other studies reported different ratios should be summarized and comparison and discussion should be provided.

The magnification bar is missing in TEM images.

Experimental section is not clear. Please better explain the synthesis and characterization.

How are the pore sizes determined?

Author Response

Response to Reviewer 1 Comments

Reviewer 2 Report

The manuscript entitled “Synthesis of PMMA microspheres with tunable diameters: evaluation as a template in the synthesis of tin oxide coatings” by Mendoza-Castellanos et al.  investigates the effect of SDS surfactant on the synthesis of PMMA with different sphere diameters.  Also, this research demonstrates that it is possible to obtain oxides coatings such as SnO2 with porosities of size-tunable using PMMA as a template. Overall, it is an interesting and well-structured manuscript that could be published to the Polymers after revision of the following points:

Please,

Abstract:

1)      Remove the first sentence as the abstract should present and introduce firstly the subject of the manuscript. The sense of this sentence is repeated in following one “This work determines that the diameters of PMMA spheres can be modified with different 23 amounts of surfactant”.

2)      Rewrite the abstract so that the objective of the manuscript to be clearer, as it is in the introduction section.

Materials and Methods:

3)      give more details on the characterization methods (section 2.2).

Results:

4)      Line 133: Refer the groups to which the characteristic peaks of PMMA are attributed. Does the amount of SDS affect the FTIR spectra?

5)      Refer if and how the different amount of surfactant affects the thermal decomposition of PMMA (section 3.2).

6)      Line 160, 215-216 : these sentences are not necessary as give the same information with the figure title. Also, add the scale values on the images of Figure 7.

7)      Line 239: Figs. 8g), not 8f)

8)      Rephrase the sentences to make sense e.g. 53-55 or 124-125. Check the whole manuscript.

Author Response

Response to Reviewer 2 Comments

Reviewer 3 Report

- The authors should justify the SnO2 coating of the microspheres (for example, any specific application for catalysis?)
- In the case of the TGA curve, the graph of the first derivative should be added to better visualize the degradation stages.
- The authors should carry out a deeper discussion about the interactions involved in stabilizing PMMA microspheres with SDS.  In its current state, it is merely descriptive
- In the Materials section, the salt precursor of SnO2, purity, and origin should be mentioned.
- A scheme about the process can help a better understanding of the readers

Author Response

Response to Reviewer 3 Comments

Reviewer 4 Report

The current work deals with The Effect of Surfactant Concentration on PMMA Sphere Diameter and Its Use as a Template for Porosity Control in SnO2 Synthesis. I appreciate the originality and relevance of your study, which examines the influence of surfactant concentration on the diameter of PMMA spheres synthesized by the emulsion polymerization method. Additionally, your work demonstrates that the size variation of PMMA spheres can be utilized as a template to synthesize SnO2 coatings with different porosities. However, I believe that your manuscript needs major revisions before it can be accepted for publication. Firstly, the abstract lacks clarity and does not fully represent the scope and significance of the study. It is suggested that you provide a brief background of the research problem and introduce the rationale behind the study. Additionally, it is essential to mention the research methods employed and the major findings in the abstract to provide readers with a comprehensive overview of the research. Secondly, the manuscript requires more detailed explanations of the experimental methods used to synthesize PMMA spheres and SnO2 coatings. It is important to provide clear descriptions of the synthesis procedure, including the amounts of reagents, surfactant, and polymer used in the emulsion polymerization process. Additionally, it would be helpful to provide more information on the parameters used to control the porosity of SnO2 coatings.

Author Response

Response to Reviewer 4 Comments
